# Optimization Processes of Clinical Chelation-Based Radiopharmaceuticals for Pathway-Directed Targeted Radionuclide Therapy in Oncology

**DOI:** 10.3390/pharmaceutics16111458

**Published:** 2024-11-15

**Authors:** Katsumi Tomiyoshi, Lydia J. Wilson, Firas Mourtada, Jennifer Sims Mourtada, Yuta Namiki, Wataru Kamata, David J. Yang, Tomio Inoue

**Affiliations:** 1Shonan Research Institute of Innovative Medicine, Shonan Kamakura General Hospital, Kamakura 247-8533, Japan; 2Department of Radiation Oncology, Thomas Jefferson University, Philadelphia, PA 19107, USA; lydia.wilson@jefferson.edu (L.J.W.); firas.mourtada@jefferson.edu (F.M.); 3Helen F. Graham Cancer Center & Research Institute, Newark, DE 19713, USA; jsimsmourtada@christianacare.org; 4Advanced Medical Center, Shonan Kamakura General Hospital, Kamakura 247-8533, Japan; y_namiki_pha_twmu@yahoo.co.jp (Y.N.); k.wataru8711@hotmail.co.jp (W.K.); dj-yang@msn.com (D.J.Y.);

**Keywords:** theranostics, ^225^Ac, ^177^Lu, imaging, dosimetry, chelation

## Abstract

Targeted radionuclide therapy (TRT) for internal pathway-directed treatment is a game changer for precision medicine. TRT improves tumor control while minimizing damage to healthy tissue and extends the survival for patients with cancer. The application of theranostic-paired TRT along with cellular phenotype and genotype correlative analysis has the potential for malignant disease management. Chelation chemistry is essential for the development of theranostic-paired radiopharmaceuticals for TRT. Among image-guided TRT, ^68^Ga and ^99m^Tc are the current standards for diagnostic radionuclides, while ^177^Lu and ^225^Ac have shown great promise for β- and α-TRT, respectively. Their long half-lives, potent radiobiology, favorable decay schemes, and ability to form stable chelation conjugates make them ideal for both manufacturing and clinical use. The current challenges include optimizing radionuclide production processes, coordinating chelation chemistry stability of theranostic-paired isotopes to reduce free daughters [this pertains to ^225^Ac daughters ^221^Fr and ^213^Bi]-induced tissue toxicity, and improving the modeling of micro dosimetry to refine dose–response evaluation. The empirical approach to TRT delivery is based on standard radionuclide administered activity levels, although clinical trials have revealed inconsistent outcomes and normal-tissue toxicities despite equivalent administered activities. This review presents the latest optimization methods for chelation-based theranostic radiopharmaceuticals, advancements in micro-dosimetry, and SPECT/CT technologies for quantifying whole-body uptake and monitoring therapeutic response as well as cytogenetic correlative analyses.

## 1. Introduction

External beam radiation therapy (EBRT) and internal targeted radionuclide therapy (TRT) are different approaches in cancer treatments. EBRT utilized intensity-modulated radiation therapy (IMRT), stereotactic body radiotherapy (SBRT), intracranial stereotactic radiosurgery (SRS), and image guidance radiation therapy (IGRT) techniques to deliver high-dose radiation to the tumor while sparing normal tissues [1]. Regardless technologic advances in EBRT enable deliver targeted radiation for the treatment of tumors, EBRT modalities do not characterize cellular targets and effects on their cellular pathway systems. Thus, EBRT may induce tissue injury when the radiation doses are high, or the large volumes of normal tissue are exposed [2,3,4,5,6,7]. One approach to establish safe radiation dose limits for healthy tissue is to incorporate the TRT of normal tissue complication probability into treatment planning. TRT evaluates the relationship between absorbed dose statistics and risk of radiation-induced tissue injury by targeting tumor heterogeneity in radionuclide distribution, pathway-directed cellular target assessment and tumor microenvironment [8,9,10,11].

Theranostic radiopharmaceuticals utilize paired radionuclides at the molecular and cellular level to precisely diagnose, stage, re-stage, and treat certain metabolic disorders and cancer. Various chelators have been reported to coordinate radionuclides along with vectors (known as chelation-conjugates) for application of theranostic concepts, including ^99m^Tc, ^68^Ga, ^61^Cu, ^89^Zr (imageable γ emitters); ^177^Lu, ^188^Re (theranostic β/γ-emitters), ^90^Y (a β emitter), ^225^Ac, and ^223^Ra (α emitters). These imageable chelation-conjugates are designed to measure changes in the target due to genomic and proteomic alterations. The application of chelation-conjugates within imaging platforms offers the potential to select patients who may respond to treatment, predict optimal treatment dosages for individual patients, particularly in cancer. Chelation-conjugates also provide the treatment option for TRT by incorporating therapeutic radionuclides when chemotherapy/EBRT and stem cell treatment fail [12,13]. Current thought is to utilize TRT at the frontline along with EBRT [14,15]. Theranostic radiopharmaceuticals using chelation-conjugate platforms are customizable and have high cellular target impact. However, TRT may be limited by the toxic effects of high-dose therapeutic radionuclides on salivary glands, kidneys, bone marrow cells, and by the intrinsic resistance of cancer cells to radiation. Thus, the optimization of safety and efficacy of TRT in cancer management includes the selection and production of radionuclides, optimization processes in radiopharmaceuticals, imaging assessment of cellular uptake, TRT distribution in cells and cytogenetic correlative analysis, and accurate radiation dosimetry. In this review article, we present on the current radiochemistry and radiation dosimetry methods. The cytogenetic analysis for radiation dose assessment, and advancements for quantifying whole-body uptake and monitoring the therapy response of chelation-based theranostic-paired radiopharmaceuticals are reviewed.

## 2. Optimization Processes of Chelation-Based Radiopharmaceuticals in Oncology

### 2.1. Instruments for Image-Guided TRT

Single photon emission computed tomography (SPECT) and positron emission tomography (PET) instruments use radiopharmaceuticals to create 3D reconstructed images of the whole body and lesions of interest. PET offers higher spatial resolution, higher sensitivity, less attenuation, and scatter artifacts than SPECT. In current clinical practice, PET and SPECT instruments are fused with CT scanners to enhance the sensitivity of these imaging methods for quantifying radiopharmaceuticals in real time. The addition of CT to PET and SPECT allows for the better delineation of tumor volumes via multiple slices by CT and serial imaging by PET and SPECT. Also, the anatomic and morphologic detail revealed by CT improves the ability to evaluate anatomic changes induced by the therapy. The cost of PET/CT is much higher than SPECT/CT. In addition, there are more radionuclides used clinically for SPECT than PET. Thus, SPECT/CT is an alternative to PET/CT.

### 2.2. Advantages of ^99m^Tc for SPECT Image-Guided TRT

Chelation-based radiopharmaceuticals provide the scientific tools for moving a concept from bench work to clinic SPECT and PET radiopharmaceuticals. They are aimed to measure the metabolic pathway-directed systems in diseases. There are two methods used in the production of radionuclides. Radionuclides produced by cyclotrons are constrained by the availability of local cyclotrons and the high cost to set up such a facility. In stark contrast with cyclotron-produced radionuclides, a generator uses a parent–daughter nuclide pair wherein a relatively long-lived parent isotope decays to a short-lived daughter radionuclide for imaging. A generator can be shipped to a clinical pharmacy and is the source from which the daughter isotope may be readily eluted. Technetium-99m (^99m^Tc) is an ideal radionuclide for diagnostic imaging studies due to its physical characteristics and low cost. It emits a 140 keV gamma ray in 89% abundance, which is commonly used by SPECT [16]. ^99m^Tc can be obtained from a ^99^Mo/^99m^Tc generator with a half-life of 6.02 h, which provides serial images for whole-body staging and re-staging applications. In addition, Na^99m^TcO_4_ (+7 the oxidation form) is a US Food and Drug Administration (FDA)-approved diagnostic imaging agent for thyroid, salivary gland, urinary bladder, gastrointestinal tract for Meckel’s diverticulum, blood pool, sentinel nodes, and nasolacrimal drainage system [16,17,18]. Despite a high radiochemical purity, and the elution of Na^99m^TcO_4_ remaining stable for several hours, Na^99m^TcO_4_ was unable to characterize cellular targets. However, ^99m^Tc in reduced forms were incorporated into various chelators along with vectors for clinical cellular imaging studies [18,19,20,21,22,23,24,25,26]. With the advantage of radiopharmaceutical grade ^99m^Tc, an active pharmaceutical ingredient (API, known as drug substance) with a reducing agent (Sn^2+^) could be formulated as a kit for clinical ^99m^Tc-chelation conjugate (drug product). These chelation-based large and small molecule imaging agents could explore the pathway-directed systems in apoptosis, angiogenesis, hypoxia, and glycosylation in cancer systems.

### 2.3. Selection of ^68^Ga for PET Image-Guided TRT

Although more SPECT radionuclides are readily available, PET offers higher spatial resolution, higher sensitivity, less attenuation, and scatter artifacts than SPECT. ^18^F-labeled (half-life 110 min) PET radiopharmaceuticals have been found useful for therapeutic monitoring chemotherapy or molecular targeted therapy in patients with varied human cancer types [27]. However, the manufacturing of ^18^F-radiopharmaceuticals requires the cyclotron and complex purification process using automated modules. Beyond cyclotron produced ^18^F, ^68^Ga (511 keV, 68 min half-life, β^+^ = 89% and EC = 11%) is a convenient alternative to ^18^F, which can be eluted from a ^68^Ge/^68^Ga generator (^68^Ge: 275-day half-life) at current good manufacturing practice (CGMP) grade. ^68^Ge/^68^Ga generators equipped with either inorganic absorbers (SnO_2_, TiO_2_, Al_2_O_3_) or synthetic resins provided a high yield of ^68^Ga and a low breakthrough of ^68^Ge [28,29]. Overall, generator-produced radionuclides reduce manufacturing time, radiation exposures, and production costs. The use of generators for clinical productions of radionuclides is shown in Table 1.

### 2.4. Selection of ^177^Lu and ^225^Ac Radionuclides for TRT

Various therapeutic radionuclides, which emitted gamma rays, X-rays, Auger electrons, α-, and β-particles, were used to kill cancer cells for cancer therapy. The radionuclides selected for TRT in cancer are based on their physical characteristics such as half-lives, production, linear energy transfer (LET), relative biological effectiveness (RBE), travel path, imaging potential, and waste disposal. Auger electrons have high LET but they have very short travel paths (2–500 nm), which limit their doses to single cells. β-particles have travel path (≤12 mm) and with low LET, which are suitable for their effectiveness in medium to large tumors. With low LET and longer travel path, β-particles cause less RBE in cancer cells and crossfire effect to normal cells. For instance, ^177^Lu (T_1/2_ = 6.647 days) has both low-energy β (0.5 MeV) and medium-energy γ (208 kev) emission, which allows theranostic approaches of cancers. Thus, ^177^Lu is safer than ^188^Re, ^90^Y, ^166^Ho, ^223^Ra, and ^153^Sm due to its theranostic capabilities. The low dose in imaging could generate radiation dosimetry for high dose ^177^Lu in treatment planning. ^177^Lu represents an ideal vehicle for selectively deposing high energies within small volumes. Its unique physical characteristics are implemented for TRT with significant results [30,31]. Conversely, α-particles have a moderate travel path (50–100 μm) and a high LET that make them especially promising to improve the tumor response relative to β-particle TRT. The high LET radiation and moderate travel path of α-particles yield a radiation-induced bystander effect whereas a β-particle yields a crossfire effect with about 500 times less cytotoxic potency and 3–5 RBE than α-particles [32,33,34,35,36,37]. Based on a review of the literature [38,39,40], ^225^Ac and ^177^Lu were promising α- and β-particles due to their unique physical characteristics and have been implemented for theranostics with significant results in the clinical management of cancer. Therefore, they are selected as representative radionuclides to be reviewed for TRT.

### 2.5. Production of ^177^Lu for TRT

There are two different reactor routes (direct and indirect) for the production of ^177^Lu [30]. The direct production route used neutron irradiation of natural lutetium (Lu) target by ^176^Lu(n,γ)^177^Lu reaction. This route represents the most inexpensive production of ^177^Lu. However, natural Lu is composed of stable isotope ^175^Lu (97.41% natural abundance) and long-lived radioisotope ^176^Lu (2.59% natural abundance with a half-life 37 billion years). Thus, the direct production of ^177^Lu results in low specific activity (20–30 Ci/mg) due to the mixture of carrier ^175/176^Lu. In addition, there is an impurity contamination with long-lived ^177m^Lu (t_1/2_ = 160.1 days) which may create safety for patients and radioactive waste disposal to the environment. The indirect production route used neutron irradiation of ytterbium (^176^Yb) target by ^176^Yb(n,γ)^177^Yb  reaction. ^177^Lu was produced after the chemical separation of the nuclear decay of ^176^Yb radionuclides. The indirect route could produce CGMP grade with no-carrier added ^177^Lu and high specific activity (>3.800 GBq/mg). In addition, there is no contamination of ^177m^Lu. This makes it possible to use high specific activity ^177^Lu for pathway-directed systems and cell surface receptor-mediated TRT.

### 2.6. Production of ^225^Ac for TRT

^225^Ac could be produced either by a generator, the cyclotron, or nuclear reactors. The advantage of using ^225^Ac from a ^229^Th/^225^Ac generator is to produce pure carrier-free ^225^Ac. The potential disadvantage of this production method is limited to small scale ^225^Ac (100–150 μCi) due to the limitation of ^229^Th amount. The production of ^225^Ac by cyclotron irradiation of ^226^Ra showed high-yield and cost-effectivity for ^225^Ac [30,31,32,33,34,35,36,37,38,39,40,41,42]. The disadvantage of these methods was coproduced by the impurity of long-lived ^227^Ac (t_1/2_ = 21.8 years), ^226^Ac (t_1/2_ = 29 h), and ^224^Ac (t_1/2_ = 2.8 h) [38,43,44]. Irradiating ^232^Th with high-energy protons has been demonstrated to produce several GBq of ^225^Ac using an intense proton beam irradiation for 10 days [45]. However, this method also produces a variety of impurities that must be removed by chemical separation and significant amounts of long-lived ^227^Ac (0.1–0.2% at end of bombardment) [46,47]. The methods for the production of ^177^Lu and ^225^Ac are summarized in Table 2.

### 2.7. Quality Assurance of ^225^Ac for TRT

Quality assurance of radionuclides optimization processes includes radionuclide identity and purity. Radionuclide identity and purity could be determined by gamma spectroscopy equipped with a high-purity germanium (HPGe) semiconductor detector, a multi-channel analyzer (MCA), and a computer-based acquisition and analysis system. This HPGe has an 80% relative efficiency, 1.8 keV FWHM at 1.33 MeV, and an energy range of 100 keV–10 MeV. A NIST-traceable dose calibrator could determine the amount of radioactivity. The radioactivity decay of the radionuclide over ten half-lives period helps to determine whether there is a breakthrough of parent radionuclide. Radionuclide purity could also be assessed by chromatographic techniques such as instant thin-layer chromatography (ITLC) and high-performance liquid chromatography (HPLC). The quality assurance of ^225^Ac and other impurities in a radioactive drug product must be identified [48,49]. The quality assurance includes the stability and establishment of ^227^Ac content limits at the time of use as ^225^Ac acceptance criteria (i.e., less than 0.3% ^227^Ac) that ^227^Ac impurity will not affect dosimetry of ^225^Ac product and may be excreted from the body with the elimination of ^225^Ac-radiopharmaceuticals if ^227^Ac remains in the ^225^Ac-radiopharmaceuticals. Jiang Z et al. reported that the accelerator produced ^225^Ac using ^232^Th target had 0.1–0.3% ^227^Ac impurity at the end of irradiation. Their biodistribution, dosimetry, toxicity studies showed no marked difference between accelerator-produced ^225^Ac and ^229^Th-derived ^225^Ac [50].

### 2.8. Chelation Chemistry in TRT

^99m^Tc and ^68^Ga paired with ^177^Lu and ^225^Ac are frequently used for chelator-based theranostic studies. ^68^Ga is with +3 charge in an acetate or chloride form. A commonly seen reduced form of ^99m^Tc is with +5 charge in an oxide form, whereas ^177^Lu and ^225^Ac are with +3 charges in nitrate or chloride forms. Diethylenetriaminepentaaceticacid (DTPA), ethylenedicysteine (EC), mercaptoacetyltriglycine (MAG3), and 1,4,7,10-tetraazacyclododecane-tetraacetic acid (DOTA) are commonly used chelators coordinated with radionuclides to fulfill the theranostic concepts [40,51]. Incorporating an imageable radionuclide in chelation-conjugate could monitor pathway-directed systems treatment outcome. Common radionuclides for imaging are ^99m^Tc, ^68^Ga, ^61^Cu, and ^89^Zr; for therapeutics are ^177^Lu and ^188^Re (β/γ-emitters), ^90^Y (a β-emitter), ^225^Ac and ^223^Ra (α-emitters). The challenges of chelation conjugates involve in the stability and validation of the mechanisms of actions.

### 2.9. Quality Assurance of Chelation Chemistry for TRT

The quality assurance of chelation-conjugates includes clinical grade unlabeled API (drug substance) and radiolabeled chelation-conjugate (drug product). For drug substance, it must fulfill conventional chemistry, manufacturing, and control (CMC) requirements. The CMC section of API includes chemical identity and purity determined by NMR, IR, mass spectrometry, elemental analysis, TLC, and HPLC. The residual solvents are determined by a gas chromatography system with flame ionization detection. For drug product, it must fulfill the appearance of the product in solutions free from particulate matter, radiochemical purity, identity, and yield assessed by radio-TLC and HPLC equipped with a radioactive detector; radiochemical stability in serum and saline assessed by radio-TLC and radio-HPLC up to 2–3 half-lives; the pH within the range 5.5–8.0; and specific activity (Ci/μmol) whereas Ci/μmol is micro molar activity assessed by NIST-traceable dose calibrator and radio-HPLC. In addition to chemical analysis, the drug product meets the safety limits of sterility and pyrogenicity assessed by bacterial endotoxins test (BET).

### 2.10. Optimization of Clinical Receptor Pathway-Directed Theranostic Radiopharmaceuticals

Macrocyclic chelators are generally more resistant to proteolysis and have a higher stability to penetrate cells to interact cellular targets than linear peptides [51]. DOTA, a hydrophilic macrocyclic chelator, has been clinically used as a stable chelator for ^99m^Tc, ^68^Ga, ^177^Lu, ^213^Bi, and ^225^Ac-labeled-chelation conjugates [52,53,54,55,56,57]. ^99m^Tc- and ^68^Ga-labeled-DOTA-conjugates provide image-guided biopsy and optimal therapeutic outcome. With the same molecular and cellular targets in DOTA-conjugates, the imaging radionuclide is switched to a theranostic paired radionuclide (^177^Lu or ^225^Ac) for TRT in cancer. Here, we review current clinical studies of theranostic-paired receptor-mediated peptide radiopharmaceuticals in fibroblast activation protein (FAP), prostate-specific membrane antigen (PSMA), and DOTA-octreotide (DOTATATE) for somatostatin receptor (SSTR) systems in neuroendocrine tumors (NETs). The structures of clinically used theranostic medicine DOTATATE, PSMA, and FAP-2286 are shown in Figure 1.

#### 2.10.1. Optimization of FAP Receptor Pathway-Directed Radiopharmaceuticals

The application of FAP for the theranostic approach in cancer has been reviewed [58]. Several FAP and FAPIs with different chelators (FAPI-2, FAPI-42, FAPI-46, FAPI-52, FAPI-72, FAPI-73, FAPI-74, FAPI-75, FAPI-76, FAP-2286, PNT-2004, NTI-1309, BIBH1), albumin-FAPI, and PEG-DOTA-FAPI were synthesized [58,59,60,61,62,63,64]. The selected structures of FAPI-04, FAPI-46 and FAP-2286 are shown in Figure 1. Unlike FAPI-04 and FAPI-46, FAP-2286 consists of two functional elements: a targeting peptide that binds to FAP and a site that can be used to attach radioactive isotopes for imaging and therapeutic use. These conjugates were labeled with various isotopes (^68^Ga, ^99m^Tc, ^177^Lu, ^225^Ac, ^90^Y, ^64^Cu, ^18^F, ^131^I) for theranostic applications. For instance, Watabe T et al., in 2020 reported the effectiveness of ^225^Ac-labeled FAPI-04 in pancreatic cancer xenograft mouse models [62]. Others reported that the treatment effects of [^177^Lu]Lu-FAPI-46 were relatively slow but lasted longer than those of [^225^Ac]Ac-FAPI-46 [63]. Of the synthesized FAP molecules, FAP-2286 was identified as a promising tracer for clinical Phase I studies because FAP-2286 maintained higher tumor selectivity and efficacy in comparison to FAPI-46 in pre-clinical models [64,65]. The clinical data revealed that [^177^Lu]Lu-FAP-2286 was relatively well tolerated, with acceptable side effects, and demonstrated long retention of the radiopeptide [65]. Prospective clinical trials using [^177^Lu]Lu-FAP-2286 in phase 2–3 and [^225^Ac]Ac-FAP-2286 for first-in-human trials are warranted.

#### 2.10.2. Optimization of PSMA Receptor Pathway-Directed Radiopharmaceuticals

[^177^Lu]Lu-PSMA-617 and [^225^Ac]Ac-PSMA-617 are commonly used in the treatment of PSMA positive metastatic castration-resistant prostate cancer (mCRPC) [66,67,68,69,70,71,72,73,74,75]. Circulating tumor cell (CTC) molecular analyses were reported to correlate to the biomarkers of [^177^Lu]Lu-PSMA-617 treatment efficacy [76,77] in mCRPC patients. Though [^177^Lu]Lu-PSMA-617 and [^225^Ac]Ac-PSMA were effective in cancer management [78,79,80,81], others reported dose-related xerostomia and dysphagia induced by [^177^Lu]Lu- and [^225^Ac]Ac-PSMA [82,83,84]. Anemia, neutropenia, myelodysplasia, bone marrow failure, acute leukemia, and chronic thrombocytopenia were treatment-induced off-target related hematologic toxicities; however, it was rare in^177^Lu and ^225^Ac-labeled PSMA treatment. The common side effects include nephrotoxicity and xerostomia. A comparison of the efficacy and toxicity in [^177^Lu]Lu- and [^225^Ac]Ac-labeled PSMA for TRT in prostate cancer is summarized in Table 3.

#### 2.10.3. Optimization of Neuroendocrine Receptor Pathway-Directed Radiopharmaceuticals

Somatostatin receptor (SSTR-2) systems are up-regulated in neuroendocrine cancers (NETs) including carcinoids [85]. The early detection of NETs allows for potentially curative treatment and the five-year survival rate for patients is less than 80% due to recurrence. The SSTR-2 binding affinity of [^68^Ga]Ga-DOTATATE was approximately 10-fold higher than that of [^68^Ga]Ga-DOTATOC [86]. However, the theranostic paired [^68^Ga]Ga-DOTATOC and [^177^Lu]Lu-DOTATOC or [^68^Ga]Ga-DOTATATE and [^177^Lu]Lu-DOTATATE showed no significant difference in providing predictive and prognostic values in progressive NET patients [87]. [^225^Ac]Ac-DOTATATE and [^177^Lu]Lu-DOTATATE were reported to be safe and effective for treating advanced and refractory NETs. [^225^Ac]Ac-DOTATATE was considered an alternative after the failure of [^177^Lu]Lu-DOTATATE treatment [88,89,90,91,92,93].

### 2.11. Pathology and Cytogenetic Correlative Analyses of Radiation Dose

α-therapy may cause double-stranded DNA breaks in cancer cells and normal cells, leading to cell death whereas β-therapy may cause single-stranded DNA damages. Cytogenetic analysis of an individual exposed to ionizing radiation has been used for the safety and efficacy of dosimetry [94]. The structural abnormalities induced by ionizing radiation in chromosomes include duplication, deletion, balanced or unbalanced translocations, inversion, and insertion. Cytogenetic testing such as karyotyping (metaphase, interphase), fluorescence in situ hybridization (FISH), chromosomal microarray analysis subtypes (CMA), and sequence (PCR, NGS) are performed in solid tumors, hematologic malignancies, and congenital diseases during chemoradiation therapy. Cytogenetic and pathology lab analyses are able to correlate to phenotype and genotype target expressions and risk of radiation-induced tissue injury for treatment planning in malignancies as well as determine appropriate therapy for prognostic stratification [95,96,97,98,99,100]. Pathology and cytogenetic techniques commonly used in characterizing tumors are summarized in Table 4.

Overall, cytogenetic testing and pathological labs provide correct diagnosis, prognostic evaluation, monitoring progression of hematologic disorders and solid tumors, improving therapeutic options (appropriate treatment regimens, response to treatment), and reducing adverse events, which are beneficial to cancer patients. Pathology and cytogenetic analyses are innovative approaches to tailor disease prevention and treatment, assessing the differences in genes, proteomics, and environments.

## 3. Radiation Dosimetry for TRT in Oncology

### 3.1. Biological Data Collection in Radiation Dose Calculations

There are three types of dose expressions in radiological protection: (1) absorbed dose (milligrays, mGy), (2) equivalent dose (millisieverts, mSv), and (3) effective dose (millisieverts, mSv). Absorbed dose is the primary measure of dose defined for TRT in oncology and represents the amount of energy deposited by radiation per unit mass. Equivalent dose represents the absorbed dose adjusted for the effectiveness of the type of radiation for causing radiation damage and effective dose includes an additional adjustment for the sensitivity of the organ or tissue to radiation damage. The adjustment factors for neither equivalent nor effective dose are well defined for the types and energy of radiation and dose rates involved in TRT. Determining absorbed dose for TRT requires biokinetic data representing the amount of radionuclide in an organ or tissue over time, most often in the form of time-activity curves. Time-activity curves can be estimated from the serial imaging of radioactivity following injection. More practical methods for estimating time-activity curves from a single post-injection measurement exist, although involve considerable uncertainty related to patient-specific biokinetics [101]. For radiopharmaceuticals with imageable photon emissions, direct imaging of the radiopharmaceutical can be achieved via SPECT. Radiopharmaceuticals leveraging radionuclides that lack a readily imageable emission (e.g., ^225^Ac) rely on a non-therapuetic radiotheranostic-paired surrogate radionuclide (e.g., ^68^Ga) with PET.

### 3.2. Software in Radiation Dose Calculations

The two principal methods of calculating absorbed dose from time-activity curves are empirical, S-value-based and mechanistic, Monte-Carlo (MC)-based algorithms. Calculations using either type of algorithm can represent organ-level dosimetry or voxel-level dosimetry. Organ-level dosimetry provides an estimate of average dose deposition in pre-defined organs and tissues throughout the body. Voxel-level dosimetry enables the visualization of spatial dose distributions and dose-volume histograms (DVHs). Medical Internal Radiation Dose (MIRD, MIRDsoft.org) is the foundational S-value-based algorithm for organ-level dosimetry [102,103]. The free suite of software applications represent a community tool developed and distributed in Microsoft Excel for calculating radiopharmaceutical dosimetry based on the MIRD schema. Two applications, MIRDfit and MIRDcalc, are particularly relevant to TRT in oncology. MIRDfit supports fitting the biokinetic data required for determining TRT doses by estimating models for time-activity curves. The MIRDcalc software includes data for 333 isotopes [104] and performs dose calculations in SI units of mGy/MBq based on anatomy selected from a library of 12 ICRP reference phantoms. Other commercial applications are also available to calculate organ-level dosimetry based on the MIRD schema (e.g., OLINDA, Hermes Medical Solutions, Stockholm, Sweeden).

### 3.3. Methods in Radiation Dose Calculations

Dosimetry and modeling of α particles in TRT have been reviewed [105,106,107,108,109,110,111,112,113,114,115]. Micro dosimetry measured by using digital autoradiographic techniques has been reported [107,108]. Voxel-level, S-value-based dosimetry solutions typically leverage Voxel S-Value (VSV) convolution with patient-specific anatomy and activity measures. Patient-specific anatomy is most often supplied by computed tomographic (CT) imaging and activity measures by post-injection quantitative SPECT imaging. Quantitative SPECT imaging gives the radionuclide activity in each voxel (i.e., 3-dimensional pixel) of the image in units of Bq/mL. The VSV kernels represent the dose distribution around a given point of radiopharmaceutical activity and are typically determined via MC simulation in reference conditions. Although developing a VSV convolution kernel for a particular radiopharmaceutical/radioisotope requires MC simulation, the subsequent application of the kernel to estimate patient-specific dosimetry is empirical in nature. Voxel-level S-value based algorithms are considered more accurate than organ-level dosimetry because of their ability to capture inhomogeneous distributions of radioactivity and tissue types, as well as patient-specific organ and lesion geometries. An example of commercial software based on VSV convolution is SurePlan MRT (MIM Software Inc., Beachwood, OH, USA. https://www.mimsoftware.com, accessed on 22 October 2024). Voxel-level MC calculations perform full MC radiation transport calculations in patient-specific anatomy (supplied by CT), based on patient-specific radionuclide activity measures (supplied by SPECT). The computational power required for MC simulation typically requires GPU-accelerated computing capabilities. MC algorithms are considered more accurate than S-value-based dosimetry because of their comprehensive modeling of individual decay emissions and transport through patient-specific tissue geometries. An example of commercial software offering a MC dosimetry solution is Torch (Voximetry Inc., Madison, WI, USA, https://voximetry.com, accessed on 22 October 2024).

## 4. Regulatory Approval Process for Chelation-Based Radiopharmaceuticals

In terms of the regulatory requirements in the US, the optimization of chelator-based theranostic radiopharmaceutical includes detailed in the US FDA CMC section of identity and purity in radionuclide production (including the method of the detection of decay daughter products), the structure characterization of unlabeled (drug substance) and the radiopharmaceutical (drug product) and biologic validation of safety, efficacy, and toxicity in preclinical models as recommended by US FDA. Preclinical studies include biodistribution with radioactive dosage escalations, blood chemistry, pathology, and cytogenetic testing of tissue DNA activities. Radiation dosimetry could be calculated by either conventional MIRD or Monte Carol methods. Correlate cytogenetic and pathologic dose response related analyses could support safety and toxicity in chelation-based theranostic-paired radiopharmaceuticals. To expedite the regulatory approval of radiolabeled chelation-conjugates in clinical imaging studies, the exploratory Investigational New Drug (IND) offers a fast-track initiative for the flexibility in the amount of pre-clinical data that need to be submitted with a traditional IND application. An exploratory IND involves very limited human exposure, which is suitable for screening, pharmacokinetic, and microdose studies. The duration of dosing in an exploratory IND study is expected to be limited (e.g., 7 days). Exploratory IND studies present fewer potential risks than do traditional phase 1 studies that look for dose-limiting toxicities, such limited exploratory IND investigations in humans can be initiated with less, or different, preclinical support than is required for traditional IND studies. Thus, this route helps to determine the selection of the candidate with different composition in radionuclides and chelators in chelation conjugates for its pharmacokinetics/pharmacodynamics in humans. Exploratory IND is not for a therapeutic study; however, it serves the basis for a further traditional phase 1 imaging study and a theranostic concept phase 1 TRT study.

## 5. Discussion

The identification of tumor-specific immunophenotyping and genetic markers has added a new dimension to the formulation of diagnosis pathway-directed systems and effective therapeutic targets. The advantage of chelation-conjugates is that they can chelate various radionuclides (^99m^Tc, ^68^Ga, ^61^Cu, ^188^Re, ^177^Lu, ^90^Y, ^111^In, ^225^Ac, Pt, etc.) for customized medication. For the selection of ^99m^Tc for image-guided TRT, the kit process is easier than complex chemistry using an automated module. ^99m^Tc (free form), eluted from a generator, is already an approved radiopharmaceutical by FDA. A kit-based ^99m^Tc-drug product by the shake-bake-shoot processes is designed for monitoring and predicting the patient selection for an optimal outcome. Other radionuclides, not approved as radiopharmaceuticals by FDA, would require automated modules to produce radiopharmaceuticals. Thus, the production and quality assurance of ^68^Ga, ^177^Lu and ^225^Ac must meet acceptable criteria. The stability and toxicity profiles of a drug product must be conducted in pre-clinical models. For instance, the unbound ^99m^Tc in ^99m^Tc-drug product is distributed in the salivary gland, thyroid, and stomach organs. The unbound ^68^Ga and ^177^Lu accumulated in salivary gland, liver, and kidneys. The recoil energy caused by the decay of α-emitters invariably destroys α-emitter-targeting vector chemical bonds, often releasing α-emitting progeny with different chelators that can lead to undesirable toxicities. Free ^225^Ac-acetate accumulates primarily in the liver and bone. The ^225^Ac daughters ^221^Fr and ^213^Bi are accumulated in the kidneys and urine. Thus, the formation of a stable ^225^Ac-chelation complex is the primary focus in therapeutic radiopharmaceuticals. To optimize the chelation conjugate in clinical applications, the CMC and quality assurances of both drug substance and drug product must meet regulatory requirements. ^187^Re, ^69^Ga, ^175^Lu, and ^139^La may be selected to incorporate in chelation-conjugates as reference standards for labeled pharmaceuticals.

^225^Ac has two principal gamma emissions at 218 keV and 440 keV from its daughters ^221^Fr and ^213^Bi, respectively. Use of SPECT/CT clinical scanners (GE Discovery NM/CT 670, GE Optima NM/CT 640, and the Siemens Symbia T6) and acquisition protocols specifically developed for α-particles such as ^223^Ra, ^225^Ac, and ^227^Th have recently been validated for [116]. Error in SPECT/CT activity quantification was on the order of 3.5% in a 3D-printed tissue phantom, making this approach acceptable for direct α-particle dosimetry in humans. Others investigated the use of ^226^Ac as a candidate isotope for in vivo imaging of preclinical validation of ^225^Ac radiopharmaceuticals, given that ^226^Ac has two gamma emissions of 158 keV and 230 keV [117].

Three clinical receptor-mediated pathway-directed theranostic radiopharmaceuticals were reviewed. They are in the field of FAP, PSMA, and SSTR. Although the efficacies of these radiopharmaceuticals were impressive, there were still poor responses in nearly one-third of patients. Here, we provided the possible mechanistic explanation along with references. Receptors are protein molecules in the target cell surface that bind ligands. Each cell-surface receptor has three main components: an external ligand-binding domain (extracellular domain), a hydrophobic membrane-spanning region, and an intracellular domain inside the cell. There are various types of cell-surface receptors. For instance, a ligand binds to enzyme-linked receptors or G-protein-linked receptors could be internalized and bound to proteins that act as regulators of mRNA synthesis to mediate gene expression. A ligand could also bind to ionic channel-linked receptors, a constitutively opened channel, and cause a conformational change in the protein’s structure that allows ions such as sodium, calcium, magnesium, and hydrogen to pass. The ligand–receptor complexes in these two types of receptors provide signal transduction and transcriptional events in the nucleus, bind to specific regulatory regions of the chromosomal DNA, and promote the initiation of transcription. However, some cell surface receptors such as immunoglobulin-like adhesion molecules, integrins, cadherins, and selectins that their primary functions are for adhesion and do not directly initiate significant internal cellular signals [118]. Others reported that the signal may not bind to its receptor due to a missense mutation in the gene sequence [119]. Thus, a ligand may bind to cell-surface receptors but does not functionally involve in the signaling to achieve cellular responses. In receptor–ligand binding assays, receptor densities (Bmax) are the summation of all-type receptors, which do not differentiate functional receptors. This may explain why one-third of patients had a poor response in receptor therapy. In addition, spare receptors, also known as receptor reserve, are receptors that remain unbound when a ligand produces a maximum response in a cell. This means that a ligand does not need to occupy all of its receptors to achieve a maximum effect from a ligand [120]. In both scenarios, there are risks in normal healthy tissue injuries due to poor response and high dosages.. For instance, these clinical theranostic radiopharmaceuticals were also accumulated at salivary glands vis a vis non-specific sodium iodide symporters. Therefore, the challenges in receptor theranostic approaches rely on (1) high specific activity of radiopharmaceuticals (prefer 1 Ci/μmol), and (2) high tumor uptake (specificity) and tumor-to-background (sensitivity) to avoid radiation induced tissue injury. Molecular and cellular imaging studies help to identify functional receptors by quantifying cell uptake in time activity curves. Transporter-based radiopharmaceuticals are alternatives to cell-surface receptor radiopharmaceuticals due to their faster intracellular uptake through over-expressed transporters in the tumors.

Cytogenetic and molecular assays could identify tumor-specific proteomic and genetic biomarkers and determine their risks and efficacy in ionizing radiation dose responses to specific therapeutic targets. The common genetic approaches used to identify tumor-specific biomarkers are conventional cytogenetic, molecular cytogenetic (fluorescence in situ hybridization, FISH), reverse transcription–polymerase chain reaction (RT-PCR), chromosomal microarray, and sequence analyses. The advantage of using cytogenetic techniques is that relevant oncogenes and tumor suppressor genes, new gene constructs, and their protein products resulting from translocations during tumor genesis as well as DNA damages could be identified and localized before, during, and after TRT. Molecular cytogenetic assays provide results when (1) tissue is insufficient or unsatisfactory for cytogenetic analysis, (2) conventional cytogenetics has failed to yield results, or (3) cryptic rearrangements are present. The pitfall for cytogenetic techniques involves in (1) the heterogeneity of tumors with the multiphasic changes (stage, re-stage) in genome mutations during progression may alter existing standard methods of data analysis; (2) the methods of DNA extraction/bioinformatic analysis, reproducibility, and the quality of the results are not standardized; and (3) they are for molecular and genetic target approaches, not tools for screening. The number of commercially available probes is limited; and (4) the use of fluorescence microscopy, thus, interpretation may be challenging when analyzing suboptimal specimens [121]. In addition, karyotyping, a sensitive phenotyping assay, may miss specific cryptic chromosome translocation. Genotyping assays (PCR, FISH, CMA) are specific for chromosomal abnormalities but not sensitive. Peter R et al. reported that the use of digital autoradiograms across whole tissues to generate 3D dose volumes and used them to evaluate the simultaneous tumor control and regional kidney micro dosimetry of a novel [^225^Ac]Ac-radiopharmaceutical for prostate cancer [122]. Therefore, integrative analyses among cytogenetics, pathology labs, and imaging assessment could enrich accurate diagnosis, prognostic evaluation, monitoring of and treatment outcome by TRT (appropriate treatment regimens, response to treatment) in targeted patient populations.

The optimization processes in theranostic radiopharmaceuticals derived from chelation-conjugate platforms could benefit patients in the (1) safety by determination of chemical and radiochemical purity and identity in CMC, the appropriate radiation dosage regimen for optimal response, and guide clinicians to determine whether resistance occurs in late stage and reduce adverse event by switching to different drug targets; (2) effectiveness by providing clinicians choices to select patients for TRT with unresectable locally advanced or metastatic cancer, and correlate to tumor specimen genomic/proteomic expressions; and (3) improving overall respond rate (ORR), overall survival (OS) and prolonging progression-free survival (PFS).

## 6. Emergent Trends

Despite the ongoing challenges of the predictive dosages for optimal response in cancer, advancing innovation and effectiveness with improved methods and new targets have continued to encourage TRT using theranostic technologies to march forward. Theranostic medicine is to reduce drug risk, optimize benefit, and lower the cost for patients by using chelation technology-driven products. Characterizing tumor pathway-directed targets using the diverse strength chelation technology enables the delivery of the next wave of innovative drugs for differential diagnosis and predictable differential response. ^161^Tb has similar chemical properties and half-life to ^177^Lu. ^161^Tb decays to ^161^Dy, with a half-life of 6.95 days. It emits β^−^ particles (154 keV) as well as γ-ray (49 keV, 75 keV that can be used for Auger electron therapy and SPECT imaging [123]. The evaluation of the safety and efficacy of ^161^Tb-labeled DOTATOC, PSMA, and FAP inhibitors for theranostic applications has been initiated [123,124,125,126]. The results of the substitution of ^177^Lu with ^161^Tb for both DOTATATE and PSMA-617 in clinical trials showed an increase in the delivered dose per unit of activity to tumor tissue by 40% [125]. This may make ^161^Tb an alternative to ^177^Lu for more effective use by TRT in small lesions and metastasis. The emergence of cytogenetic and pathological analyses has provided us with insights into system biology and bioinformatics linking to digital health systems using artificial intelligence (AI) and genomic risk factors for chromosomal abnormalities, which are essential to the assessment of radiation dose safety and efficacy in cancer patients. Emergent trends on how optimization processes of chelation-based radiopharmaceuticals for pathway-directed systems impact healthcare systems are shown in Figure 2. Thus, chelation-conjugates could improve the ORR, OS, and PFS of patients with cancer. The emerging trends in chelation technologies (horizon technologies, government/field initiatives for fast-track clinical protocol approval) will change the trajectory of human health by product portfolios to meet critical unmet needs that our academic field and industry is witnessing.

## 7. Conclusions

Molecular imaging techniques play a critical role in the development of novel therapies since they provide information on target expressions, pathway activities, and cell functions in the intact organism. The chelation-conjugate technology platform could transform a diagnostic imaging drug to a predictive radiotherapeutic drug, the key theranostic concept for precision medicine. The imaging component advances understanding pathway-directed systems using first-in-class chelation-conjugates to deliver lethal radiation dose to modulate RNA/DNA proliferation in cancer. The factors influence the accuracy of absorbed dose determination include the distribution of the radionuclide within the body and its heterogeneous distribution of the radionuclide within the critical organ may vary with patients’ age, disease, or pathological conditions. Recent advancements in quantitative SPECT/CT and radiation transport methods for TRT treatment planning are paving the way to make this a reality for cancer management similar to EBRT. Overall, the optimization of TRT in cancer management relies on the integration of (1) radiochemistry processes in the production and quality assurance of radionuclides, CMC of synthetic chelation conjugates and quality assurance of labeled of radiopharmaceuticals; (2) pharmacology and toxicology using in vivo and ex vivo imaging assessment of cell uptake and TRT distribution in cell nuclei; and (3) the standardization of radiation dosimetry determination on the micro level to better correlate to immunohistochemistry, proteomic, and cytogenetic analyses.

## Figures and Tables

**Figure 1 pharmaceutics-16-01458-f001:**
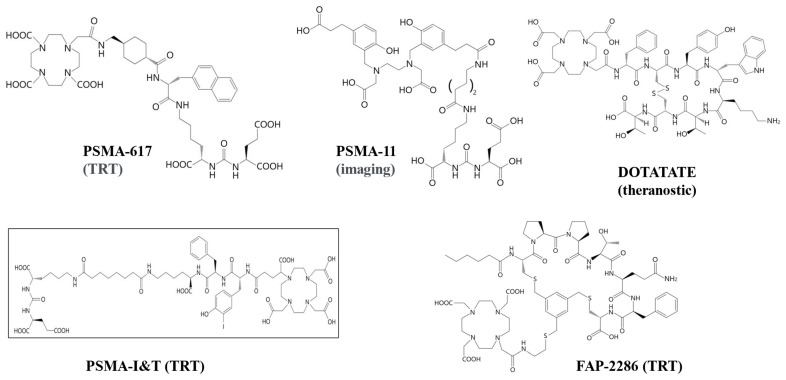
Structures of clinical theranostics: DOTATATE, PSMA and FAP-2286.

**Figure 2 pharmaceutics-16-01458-f002:**
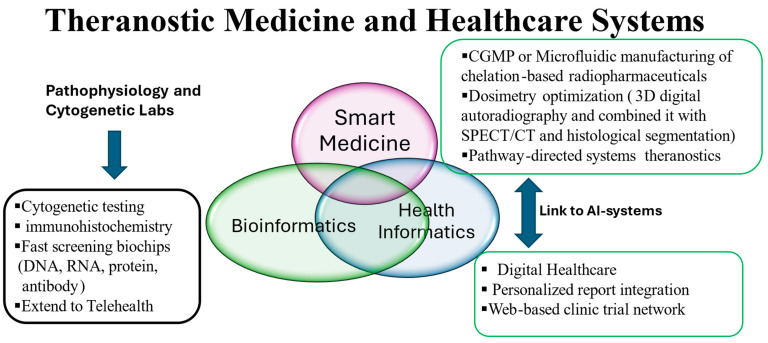
Emergent trends on how optimization processes of chelation-based radiopharmaceuticals for pathway-directed systems impact healthcare systems.

**Table 1 pharmaceutics-16-01458-t001:** Generators for clinical productions of radionuclides.

Generator	Parent Half-Life	Daughter Half-Life	Frequency of Milking
^99^Mo/^99m^Tc	2.8 days	6 h	*
^188^W/^188^Re	70 days	17 h	24 h
^68^Ge/^68^Ga	271 days	68 min	6 h
^229^Th/^225^Ac	7920 years	9.9 days	*

*: Frequency of milking depends upon generator size.

**Table 2 pharmaceutics-16-01458-t002:** Production methods of ^177^Lu and ^225^Ac for TRT.

Method	Advantage	Disadvantage
Direct production route. Neutron irradiation of natural lutetium (^175^Lu, 97.41%; ^176^Lu, 2.59% ) target by ^176^Lu(n,γ)^177^Lu reaction.	Produce large quantity and inexpensive production of ^177^Lu.	Produce low specific activity ^177^Lu (20–30 Ci/mg) and an impurity ^177m^Lu (t_1/2_ = 160.1 days), which may create safety and radioactive waste disposal concerns.
Indirect production route. Neutron irradiation of ytterbium (^176^Yb) target by ^176^Yb(n,γ)^177^Yb reaction.	Produce CGMP grade with no-carrier added ^177^Lu and high specific activity (>3.800 GBq/mg). No contamination of ^177m^Lu.	^177^Lu was produced after chemical separation of the nuclear decay of ^176^Yb radionuclides.
^229^Th/^225^Ac generator	Produce pure carrier-free ^225^Ac and free of other Ac-isotopes. The generator has half-life (t_1/2_ = 7.3 years).	Limit to small scale ^225^Ac (100–150 mCi) due to the limitation of ^229^Th amount.
Accelerator: irradiation of ^226^Ra via ^226^Ra(p,2n)^225^Ac, ^226^Ra(γ,n)^225^Ra, ^226^Ra(3n,γ)^229^Ra, ^226^Ra(p,n)^226^Ac reactions.	Produce large amount of ^225^Ac year around.	Contain impurity of ^227^Ac (t_1/2_ = 21.8 years) and other Ac-isotopes (^226^Ac t_1/2_ = 29 h; ^224^Ac t_1/2_ = 2.8 h).

**Table 3 pharmaceutics-16-01458-t003:** Comparison of efficacy and toxicity in [^177^Lu]Lu and [^225^Ac]Ac-labeled PSMA for TRT in mCRPC * patients.

[^177^Lu]Lu-PSMA	[^225^Ac]Ac-PSMA
β-emitter with g 113, 208 keV LET 0.5 MeV, 0.2 keV/μm Range 2 mm (crossfire effect) Half-life 6.7 days >50% PSA decline rate of 46% ** Adverse event: resistance, sialadenitis, less nephrotoxicity	α-emitter with 4a, and g 440 keV LET 5.8 MeV, 100 keV/μm Range 40–100 μm (by-stander effect) Half-life 9.9 days >50% PSA decline rate of 60% ** Adverse event: sialadenitis, more nephrotoxicity

* mCRPC: metastatic castration resistant prostate cancer. ** References: [80,81,82].

**Table 4 pharmaceutics-16-01458-t004:** Pathology and cytogenetic techniques in characterizing tumors (detect chromosome deletions, duplications, translocations, insertions, and inversions).

Pathology Lab	Cytogenetic Profiling
Immunohistochemistry (IHC) staining and in situ hybridization (ISH) Molecular diagnostics, including RT-PCR and comparative genomic hybridization (CGH) Karyotyping, spectral karyotyping (SKY) Fluorescence in situ hybridization (FISH) Flow cytometry for immunophenotyping and minimal residual diseases (MRDs)	Fluorescence in situ hybridization (FISH) Reverse transcription-polymerase chain reaction (RT-PCR) analyses Reverse phase protein arrays (RPPA) for functional proteomics Chromosomal microarray analysis (CMA) Sequencing-based tests: next-generation sequence (NGS)

## Data Availability

The data and references cited in this review article were obtained from a public repository that issues datasets with DOIs or journal numbers and years where available.

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
