# Peer review of "Optimization Processes of Clinical Chelation-Based Radiopharmaceuticals for Pathway-Directed Targeted Radionuclide Therapy in Oncology"

_pharmaceutics, 2024, doi:10.3390/pharmaceutics16111458_

Round 1

Reviewer 1 Report

Comments and Suggestions for Authors

The article is a detailed, well-researched review of chelation-based radiopharmaceuticals for TRT in oncology. It effectively covers various technical and clinical aspects, although certain sections could benefit from added clarity and minor adjustments to improve flow. Emphasizing emerging strategies, including advancements in dosimetry and receptor-based targeting, would also enhance the impact of the review.

Two comments:

1. mercaptoacetyltriglycine (MEG3), 1,4,7,10-tetraazacyclododecane-tetraacetic acid (line 255): The abbrev is MAG3

2. specific activity (Ci/mmol) assessed by NIST-traceable dose calibrator and radio-HPLC (line 288): Ci/mmol is molar activity

Author Response

Comment1: mercaptoacetyltriglycine (MEG3), 1,4,7,10-tetraazacyclododecane-tetraacetic acid (line 255): The abbrev is MAG3

Response1: Thanks for the valuable comments. Per comments, we have revised the text in depth to clarify questions raised by the reviewer. For instance, the abbreviation of MEG3 has been corrected to MAG3

Comment2: specific activity (Ci/mmol) assessed by NIST-traceable dose calibrator and radio-HPLC (line 288): Ci/mmol is molar activity

Response2: The sentence “specific activity (Ci/mmol) assessed by NIST-traceable dose calibrator and radio-HPLC (line 288): Ci/mmol is molar activity” has been clarified to “specific activity (Ci/mmol) whereas Ci/mmol is micro molar activity assessed by NIST-traceable dose calibrator and radio-HPLC”.

Reviewer 2 Report

Comments and Suggestions for Authors

The review article entitled “Optimization Processes of Clinical Chelation-Based Radiopharmaceuticals for Pathway-Directed Targeted Radionuclide Therapy in Oncology” by  Katsumi Tomiyoshi, Lydia Wilson, Firas Mourtada, Jennifer Sims Mourtada , Yuta Namiki, Wataru Kamata, David J. Yang and Tomio Inoue deals with the recent state of targeted radionuclide therapy based on chelation of radionuclides.

The topic is recently very actual and so of an interest for broader scientific community. The choice of the material is well made and presented in sufficient details.

On a minus side, the narrative is on a wordy side, with lengthy paragraphs, crowded with less important details. The order of material presented seemingly do not follow any obvious structure.

A more concise narrative that follows a clear logical path, with carefully separated paragraphs (one paragraph – one topic) would greatly improve readability of the paper.

It may be advisable to separate the material in topics and sub-topics (for example topic radiopharmaceuticals, subtopics radionuclides, chelators, targets etc.), topic instrumentation, sub-topics CT, PET, SPECT etc…), topic therapy, sub-topics application, dosimetry etc… and follow that logic through the text.

Also, it may be advisable to include the radionuclides recently “in trend”, for example 161Tb, especially when dealing with FAP-based targeting, etc.

On a technical side, there are few errors caused by lack of scientific accuracy. For example, in Fig.1 title, “Strucutres of clinical theragnostics: DOTATATE, PSMA and FAP”, it is not FAP (FAP stands for Fibroblasts Activating Protein) but FAP ligand with inhibitory property, named FAP-2286 which is shown.

Comments on the Quality of English Language

The narrative is on a wordy side, with lengthy paragraphs, crowded with less important details. The order of material presented seemingly do not follow any obvious structure.

A more concise narrative that follows a clear logical path, with carefully separated paragraphs (one paragraph – one topic) would greatly improve readability of the paper.

Author Response

Comments1:  the narrative is on a wordy side, with lengthy paragraphs, crowded with less important details. The order of material presented seemingly do not follow any obvious structure.

A more concise narrative that follows a clear logical path, with carefully separated paragraphs (one paragraph – one topic) would greatly improve readability of the paper.

It may be advisable to separate the material in topics and sub-topics (for example topic radiopharmaceuticals, subtopics radionuclides, chelators, targets etc.), topic instrumentation, sub-topics CT, PET, SPECT etc…), topic therapy, sub-topics application, dosimetry etc… and follow that logic through the text.

Response1: Thanks for the valuable comments.  The text has been revised in depth to clarify questions raised by the reviewer. We also have rephrased few long sentences and reduces duplication according to the highlighted parts. We took the Reviewer-2’s comments by revising the text to be more concise narrative that follows a logical path with carefully separated paragraphs. The material in topics and sub-topics were separated. We agree that the flow would greatly improve readability of the paper.

Comments2: Also, it may be advisable to include the radionuclides recently “in trend”, for example 161Tb, especially when dealing with FAP-based targeting, etc.

Response2:  Thanks for the valuable comments. Under the section of Emergent Trend, we have included a paragraph of the evaluation of safety and efficacy of 161Tb -labeled DOTATOC, PSMA and FAP inhibitors for theranostic applications in cancer management [123-126].  The substitution of 177Lu with 161Tb for both DOTATATE and PSMA-617, results in clinical trials showed an increase in the delivered dose per unit of activity to tumor tissue by 40% [125]. This may make 161Tb an alternative to 177Lu for more effective use by TRT in small lesions and metastasis. References were included.

Comments3: On a technical side, there are few errors caused by lack of scientific accuracy. For example, in Fig.1 title, “Strucutres of clinical theragnostics: DOTATATE, PSMA and FAP”, it is not FAP (FAP stands for Fibroblasts Activating Protein) but FAP ligand with inhibitory property, named FAP-2286 which is shown.

Response3: Fig 1 title has been corrected to “Structures of clinical theranostics: DOTATATE, PSMA and FAP-2286”
